# Transcriptomic Analysis of the Response of the Toxic Dinoflagellate *Prorocentrum lima* to Phosphorous Limitation

**DOI:** 10.3390/microorganisms11092216

**Published:** 2023-08-31

**Authors:** Xiukun Wan, Ge Yao, Kang Wang, Yanli Liu, Fuli Wang, Hui Jiang

**Affiliations:** State Key Laboratory of NBC Protection for Civilian, Beijing 102205, China; xiukunwan@126.com (X.W.); bzyaoge@163.com (G.Y.); yiyongjun1949@163.com (K.W.); liuh306@hotmail.com (Y.L.); wangfuli5728@163.com (F.W.)

**Keywords:** *Prorocentrum lima*, phosphorus limitation, transcriptomics, physiological analyses, metabolic mechanisms

## Abstract

Some dinoflagellates cause harmful algal blooms, releasing toxic secondary metabolites, to the detriment of marine ecosystems and human health. Phosphorus (P) is a limiting macronutrient for dinoflagellate growth in the ocean. Previous studies have been focused on the physiological response of dinoflagellates to ambient P changes. However, the whole-genome’s molecular mechanisms are poorly understood. In this study, RNA-Seq was utilized to compare the global gene expression patterns of a marine diarrheic shellfish poisoning (DSP) toxin-producing dinoflagellate, *Prorocentrum lima*, grown in inorganic P-replete and P-deficient conditions. A total of 148 unigenes were significantly up-regulated, and 30 unigenes were down-regulated under 1/4 P-limited conditions, while 2708 unigenes were significantly up-regulated, and 284 unigenes were down-regulated under 1/16 P-limited conditions. KEGG enrichment analysis of the differentially expressed genes shows that genes related to ribosomal proteins, glycolysis, fatty acid biosynthesis, phagosome formation, and ubiquitin-mediated proteolysis are found to be up-regulated, while most of the genes related to photosynthesis are down-regulated. Further analysis shows that genes encoding P transporters, organic P utilization, and endocytosis are significantly up-regulated in the P-limited cells, indicating a strong ability of *P. lima* to utilize dissolved inorganic P as well as intracellular organic P. These transcriptomic data are further corroborated by biochemical and physiological analyses, which reveals that under P deficiency, cellular contents of starch, lipid, and toxin increase, while photosynthetic efficiency declines. Our results indicate that has *P. lima* evolved diverse strategies to acclimatize to low P environments. The accumulation of carbon sources and DSP toxins could provide protection for *P. lima* to cope with adverse environmental conditions.

## 1. Introduction

Harmful algal blooms (HABs) occur frequently on a global scale with the deterioration of the marine environment, and have severe impacts on marine ecosystems, aquaculture, and human health [1,2]. Dinoflagellate is the main factor causing HABs, and estimated to be responsible for 80% of HABs in the global ocean [3,4]. *Prorocentrum lima* is a widely distributed marine benthic dinoflagellate usually found attached to large algae, seagrasses, and substrates, and it mainly inhabits temperate, subtropical, and tropical seas [5,6,7]. As a known producer of diarrheic shellfish poisoning (DSP) toxins, such as okadaic acid (OA), dinophysistoxin-1 (DTX1), and dinophysistoxin-2 (DTX2) [5,6,7], *P. lima* is associated with DSP incidents in the world, such as UK [6], Japan [8], China [9], Canada [10], and Argentina [11]. Also, large biomass blooms of the species have been reported along the coast of India [7], Italy [12], and Turkey [13]. DSP outbreaks in coastal waters are hazardous events as they can cause seafood toxicity and affect toxins transfer through the food chain, including humans [14]. Consequently, the formation mechanisms of dinoflagellate blooms and the regulation of physiological processes have become research hotspots in the field of HABs.

Previous studies have indicated that changes in environmental factors could affect the proliferation of dinoflagellates and metabolite synthesis, as well as bloom formation [15,16,17]. Nutrients are essential components for the growth and metabolism of dinoflagellates, therefore, optimizing the supply of nutrients is crucial [17,18,19,20]. Low concentrations of nutrients in dinoflagellate cultures have been shown to increase the production of toxins [17,21,22]. Among various nutrient supplements, the macronutrient phosphorus (P) is considered as a vital component as it is involved in various biochemical processes, such as nucleic acid biosynthesis, photosynthesis, energy conservation, lipid membrane formation, the regulation of many enzymes, and signal transduction [23,24,25,26]. However, low concentrations of dissolved inorganic phosphorus (DIP) in many waters fail to meet dinoflagellate demands. To adapt to P-deficient environments, phytoplankton species have evolved diverse strategies [24,25,27,28]. For example, some species such as *Thalassiosira pseudonana*, *Skeletonema costatum*, and *Trichodesmium* spp. can absorb and store large amounts of P as polyphosphates in their vacuoles, which is used when ambient P is deficient [29,30,31], while other species, such as *Karenia mikimotoi*, *Prorocentrum donghaiense*, and *Alexandrium catenella* can improve their phosphate uptake rate, induce the expression of phosphate transporters, or utilize dissolved organic P (DOP) [28,32,33]. In addition, some species such as *Thalassiosira pseudonana*, *Chaetoceros affinis*, and *Alexandrium catenella* can lower the cellular demand of DIP by using non-phosphorus lipids in the membrane in response to phosphorus scarcity [34,35,36]. For plants, they also evolved a wide range of morphological, physiological, molecular, and symbiotic strategies to improve P uptake and homeostasis, such as modifying root architecture and morphology, the induction of acid phosphatases and recycling enzymes, increasing expression of P transporters, developing symbioses with arbuscular mycorrhizal fungi, and regulating complex P regulatory networks in plant cells [37,38,39]. Therefore, different species of phytoplankton and plants have developed various strategies to cope with phosphate stress. However, the adaptive and response mechanisms of *P. lima* to ambient P limitation remain poorly understood.

Despite tremendous progress in research on the morphological, phylogenetic, and toxic component analysis of *P. lima* [6,40,41,42], comprehensive understanding of the molecular mechanisms involved in P acclimation remains under-investigated. This study was to gain insight into the global regulation of metabolic pathways in response to P-limitation and characterize DSP toxin biosynthesis in *P. lima*. The transcriptomes of *P. lima* under P-replete and P-deficient conditions were compared and the differentially expressed genes were characterized. In addition, changes in lipid, starch, and DSP toxin contents, as well as photosynthetic parameters in *P. lima* under P-limited conditions, were examined. Results of this study contribute to explore the adaptive strategy and metabolic mechanisms of *P. lima* to ambient P deficiency.

## 2. Materials and Methods

### 2.1. Algal Culture

*Prorocentrum lima* (strain CCMA071) was kindly provided by the Center for Collections of Marine Bacteria and Phytoplankton of Xiamen University, China. Cells were cultured in filtered (0.22 μm, Millipore, Burlington, MA, USA) and sterilized seawater with a salinity of 30/1000, and enriched with f/2 media. All cultures were grown at 22 °C in an artificial illumination incubator under a day and night cycle condition (12/12 h). Cell concentration was estimated using a cell counting plate under an inverted optical microscope (CKX53, Olympus, Tokyo, Japan).

To explore metabolic mechanisms expressed under different P culture conditions, *P. lima* in an initial cell concentration of 6000 cell mL^−1^ was grown under phosphate nutrient-sufficient conditions (1P, 36 μM NaH_2_PO_4_) and P-limited conditions (1/4 P, 9 μM NaH_2_PO_4_, and 1/16 P, 2.25 μM NaH_2_PO_4_). Three replicates were grown with each treatment. Samples of each culture were collected every three days and centrifuged at 3000× *g* for 5 min at 22 °C, for cell enumeration and analysis of physiological parameters. In addition, cells from each treatment were harvested on day 30 for RNA extraction, transcriptomic sequencing, qPCR, physiological observation, toxin analysis, etc.

### 2.2. Toxin Analysis

Cells were harvested from 50 mL of cultures on day 30 and resuspended in 2 mL methanol, and crushed using an ultrasonic liquid processor (Scientz-950ED, Ningbo, China) under ice-bath conditions for 30 min. After centrifugation at 10,000× *g* (4 °C for 15 min), pellets were subject to ultrasonic crushing twice and the supernatants were combined for toxin analysis. Quantitative analysis was performed using high-performance liquid chromatography coupled to a triple quadrupole mass spectrometer (HPLC–MS/MS, Agilent 6400 QQQ MS, Santa Clara, CA, USA) as described in [2]. Dinophysistoxin-1 (DTX1), dinophysistoxin-2 (DTX2), and okadaic acid (OA) standards were purchased from the Institute for Marine Biosciences, National Research Council Canada (Ottawa, ON, Canada).

### 2.3. Analysis of Physiological Parameters

Samples of cultures were collected every three days. Cells were counted using a plankton counting chamber with counting area 20 × 20 mm under a light microscope. A total of 50 μL of culture was added to the counting chamber after thoroughly mixing, and the counting was repeated three times for each sample. Cell growth rate was estimated using an equation for (cell µ): µ = (Ln Nf − Ln N0)/(Tf − T0). µ (d − 1) is the specific growth rate; N0 and Nf are the initial and final concentrations of cells mL^−1^, and Tf − T0 is the time interval from day 0 to day f.

The photochemical efficiency of photosystem II (*Fv*/*Fm*) and instantaneous chlorophyll fluorescence (*Ft*) were measured using AquaPen AP110 (Photon Systems Instruments, Drásov, Czech Republic). Aliquots of 20 mL were centrifuged at 3000× *g* for 5 min at 22 °C, the supernatants used to determine the extracellular phosphorus concentration according to the phosphomolybdenum blue spectrophotometric method [4] and pellets resuspended in 0.017 M magnesium sulfate (MgSO_4_) and crushed using ultrasonication in ice-bath for 20 min. After centrifugation at 10,000× *g* (4 °C for 15 min), the supernatants were used to measure the cellular phosphorus content as well as the bulk alkaline phosphatase activity (APA). APA was evaluated using an assay kit (Beijing Solarbio Science & Technology Co., Ltd., Beijing, China) according to the manufacturer’s instructions. The absorbance at 492 nm was measured on a microplate reader (Multiskan FC, Thermo Scientific, Waltham, MA, USA).

### 2.4. Confocal Laser Scanning Microscope (CLSM) Analysis

Morphology of lipid droplets in *P. lima* cells was observed under a laser-scanning confocal microscope (Leica SP8 DIVE/Falcon, Wetzlar, Germany) using the Nile red staining method as described in [19]. Briefly, cells from 50 mL of algal culture under different treatments were collected by centrifugation (3000× *g* for 5 min at 22 °C) at day 30, the obtained pellets resuspended in 3 mL f/2 media (20% DMSO) for 20 min at room temperature, and 30 μL of Nile red (0.1 mg/mL in acetone) added. The morphology of the Nile-red-stained cells was observed using a laser-scanning confocal microscope with an excitation wavelength of 488 nm and emission of 528 nm. Pictures from at least 10 cells for each sample were randomly captured.

### 2.5. Lipid Analysis

Lipid content in the Nile-red-stained *P. lima* cells was analyzed as described by Hou et al. [19] and their relative fluorescence intensity was measured to compare lipids content among different treatments. Briefly, algal cells from three aliquots of 50 mL of *P. lima* cultures were collected by centrifugation (3000× *g* for 5 min at 22 °C) on day 30, the obtained pellets resuspended in 3 mL of f/2 media (20% DMSO) for 20 min at room temperature, and 30 μL of Nile red (0.1 mg/mL in acetone) added. The stained samples were vigorously shaken and incubated under darkness for 20 min at room temperature, and transferred to a 96-well plate to determine fluorescence intensity (excitation at 485 nm, emission at 538 nm) using a microplate reader (Fluoroskan™ FL, Thermo Scientific).

### 2.6. Starch Content Measurements

A total of 50 mL of *P. lima* cells from three replicates of the P-limited and control cultures was collected via centrifugation (3000× *g* for 5 min at 22 °C). Starch was extracted and detected using a starch content assay kit (Boxbio, Beijing, China) according to the manufacturer’s instructions.

### 2.7. RNA Extraction, Sequencing, and Transcriptome Analysis

Three replicates of *P. lima* cultures were collected on day 30 and washed with 1× PBS to remove possible bacterial contaminants. A TransZolTM Plant kit (TransGen Biotech, Beijing, China) was used for total RNA extraction. RNA purity was assessed with a Nanodrop Spectrophotometer (Thermo Scientific, USA), RNA quantity was assessed using a Qubit 2.0 Fluorometer (Thermo Scientific, USA), and RNA integrity was assessed by a Fragment Analyzer 5400 (Agilent, USA). Sequencing libraries were generated using NEBNext^®^ UltraTM RNA Library Prep Kit for Illumina (NEB, Ipswich, MA, USA) as described in [2]. Then, the library preparations were sequenced on an Illumina Novaseq 6000 platform and 150 bp paired-end reads were generated.

Fastp software (version 0.19.7) was used to perform basic statistics on the quality of the raw reads. After quality filtering, the raw reads were transformed into the processed reads, and the Trinity software (version 2.13.2) used to assemble the processed reads, which were used as the reference sequences for hierarchical clustering. The longest cluster sequences obtained after corset hierarchical clustering were used as unigenes for subsequent analysis.

To improve the annotations of the dinoflagellate transcriptomes, searches for all the assembled unigenes were performed in multiple databases, including non-redundant (NR) database, Gene Ontology (GO), Kyoto Encyclopedia of Gene and Genomes (KEGG) analyses, Protein family (Pfam), Clusters of Orthologous Groups of Proteins/euKaryotic Ortholog Groups (COG/KOG), Swiss-Prot, and TrEMBL databases. The gene expression levels were estimated using fragments per kb per million reads (FPKM) for each sample. Differential expression analysis (fold changes) of each group was performed using DESeq2 (version 1.22.2), and the genes with log_2_ (fold change) ≥ 1 and *p*-value < 0.05 were assigned as differentially expressed.

### 2.8. RT-qPCR

Total RNA was extracted using a TransZolTM Plant kit according to the protocol (TransGen Biotech, China). First strand cDNA was synthesized from 1 μg of total RNA using a Prime Script™ RT reagent kit (TaKaRa, Beijing, China). To confirm the RNA-Seq assay results, RT-qPCR was performed to validate the differential expression of some genes (DEGs) between the P-replete and P-limited cells. Specific primers (Appendix A) were designed using online design software Primer3Plus (version 3.3.0) (https://www.primer3plus.com/, URL (accessed on 2 March 2023)) based on the obtained transcriptome unigenes of *P. lima*. *β-tubulin* was used as an internal gene to normalize the expression of target genes.

PCR was performed in a CFX96 fluorescence quantitative PCR System (Bio-Rad, Hercules, CA, USA) using iTaq Universal SYBR^®^ Green Supermix (Bio-Rad, USA) with the following PCR reaction protocol: 95 °C for 30 s, 40 cycles at 95 °C for 5 s, 60 °C for 30 s, followed by a final melting curve from 65 °C to 95 °C for 5 s with an increment of 0.5 °C. The relative expression levels were calculated based on the 2^−ΔΔCt^ relative response method.

### 2.9. Statistical Analysis

Statistical analyses were performed using the software GraphPad Prism 5 (version 5.01). All data were expressed as mean ± standard deviation. Student’s *t*-test was used to compare parameters among the different treatments after testing for the variance homogeneity. A difference with a *p* value < 0.05 was considered statistically significant.

## 3. Results

### 3.1. Physiological Responses of P. lima under Phosphate Limitation

*P. lima* cultures with an initial concentration of 6000 cells mL^−1^ were grown with two P-limited (1/4 P and 1/16 P) and one P-replete (1P) treatment, and showed similar growth curves until day 9 when their growth curves started to diverge (Figure 1A). Cells under the P-replete treatment reached a final cell concentration of 43,200 ± 1697 cells mL^−1^ after 30 days (Figure 1A). In contrast, the maximum cell concentration for 1/4P-limited and 1/16P-limited cultures after 30 days was 29,000 ± 3111 and 16,200 ± 282 cells mL^−1^, respectively (Figure 1A). The maximum cell growth rate in P-replete and 1/4P-limited groups occurred on day 15–18, with 2350 ± 259, and 1392 ± 200 cells mL^−1^ day^−1^, respectively (Figure 1B). The average cell growth rate in P-replete, 1/4P-limited, and 1/16P-limited cultures during the 30 days cycle was 1240, 766, and 340 cells mL^−1^ day^−1^, respectively (Figure 1B).

The extracellular P concentration decreased in both P-replete and -deficient cultures in parallel with increasing cell concentration, and P concentration in P-replete cultures was undetectable on day 18 (Figure 1C). The cellular P contents increased in all the treatments as the P in the solution was absorbed rapidly by day 3, but decreased gradually after day 6 as the P concentration in the solution declined (Figure 1D). In the P-replete and 1/4P-limited treatments, the bulk APA was very low from day 1 to 12, slightly increased from day 12 to 21, and then remained at a relatively stable level (Figure 1E). Conversely, the bulk APA in the 1/16P-limited cultures began to increase from day 3, and was greatly enhanced by day 27 (Figure 1E). Overall, the bulk APA in the 1/16P-limited cells was higher than that in P-replete and 1/4P-limited cells.

Changes in *Ft* values were similar in P-replete and 1/4P-limited cells until day 21 when their curves started to diverge (Figure 1F). *Ft* values in P-replete cells reached a stationary phase on day 30–33 and a peak value around 36,711 ± 2575, while *Ft* maximum in 1/4P-limited cells (around 29,727 ± 1103) was reached on day 21 and then decreased gradually until around 23,555 ± 2143 on day 33. In the 1/16P-limited cultures, *Ft* values were significantly lower than in the other two groups, a peak (around 14,489 ± 690) was reached on day 21, and then decreased gradually until around 5526 ± 215 on day 33 (Figure 1F).

*Fv*/*Fm* increased from day 0 to 12 in both P-replete and P-limited cells (Figure 1G). For the P-replete and 1/4P-limited cells, there were similar variation trends for *Fv*/*Fm* values, which reached a maximum value (around 0.60) on day 12 and then decreased gradually until about 0.40 on day 33 (Figure 1G). In comparison, in the 1/16P-limited cultures, the similar trend in *Fv*/*Fm* changes was found from day 0 to 21 as in the P-replete and 1/4P-limited cells, but the values significantly decreased from day 24 to the end of the experiment, at around 0.20 (Figure 1G).

### 3.2. Transcriptomic Analysis

Qualified cDNA libraries of *P. lima* samples were sequenced using the Illumina Novaseq 6000 platform, resulting in an average of 112,111,739 raw reads (Table 1). After raw data filtering, 85,019,647 clean reads were acquired, and each data set of clean reads exhibited Q20 > 97% with an average GC content of 58.76%. All clean reads were submitted to the NCBI Sequence Read Archive database (BioProject ID: PRJNA966759). The clean reads were then used for assembly by Trinity, resulting in 98,860 unigenes with an N50 of 1581 nt and an average sequence length of 1119 nt.

To obtain comprehensive gene function annotations, BLASTx searches were performed in the KEGG, NR, Swiss-Prot, TrEMBL, KOG, GO, and Pfam databases (Table 1). The results identified that 17.54% of the unigenes had significant similarity in the KEGG database, 24.22% of the unigenes had BLASTx hits in the NR database, 13.74% of the unigenes had an annotated match in the Swiss-Prot database, 47.14% of the unigenes had an annotated match in the TrEMBL database, 16.95% of the unigenes had BLASTx hits in the KOG database, 36.88% of the unigenes had an annotated match in the GO database, and 44.72% of the unigenes had BLASTx hits in the Pfam database. Overall, 61.34% of the unigenes were annotated in at least one database.

### 3.3. Transcriptomic Analysis of Differential Gene Expression

Among these transcripts, 148 unigenes were significantly up-regulated, and 30 unigenes were down-regulated under 1/4 P-limited conditions (1/4 P vs. 1P, |log_2_ (fold change)| ≥ 1; *p*-value < 0.05). In addition, 2708 unigenes were significantly up-regulated, and 284 unigenes were down-regulated under 1/16 P-limited conditions (1/16 P vs. 1P, |log_2_ (fold change)| ≥ 1; *p*-value < 0.05) (Figure 2A,B). KEGG enrichment analysis of the differentially expressed genes shows that the top pathways are ribosome, metabolic pathways, biosynthesis of secondary metabolites, photosynthesis, and ubiquitin-mediated proteolysis (Figure 2C,D). The differentially expressed genes were also detected using Gene Ontology (GO) clustering (Appendix A). GO analysis includes three parts: biological process, cellular component, and molecular function. The top biological processes were metabolic processes, cellular processes, and response to stimulus. The top cellular components were cell parts, membrane parts, and organelles. The top molecular functions were binding, catalytic activity, and transporter activity. All the differentially expressed genes under P-limited conditions were listed and annotated, as shown in Appendix A. The representative differentially expressed genes between 1/16 P-limited conditions and P-replete conditions are shown in Table 2.

As many as 76 ribosomal subunit proteins were found to be up-regulated under the 1/16 P-limited conditions, such as the small subunit ribosomal proteins S20e, S11e, S5e, S4e, and S23e, and the large subunit ribosomal proteins L34e, L10e, L13e, and L14e. The stress–shock proteins, such as Hsp 70, Hsp 90, Hsp 83, and HSPA1s, were significantly up-regulated. Genes related to glycolysis, gluconeogenesis, and oxidative phosphorylation such as pyruvate decarboxylase (cluster-1527.74515), phosphoglycerate kinase (cluster-1527.83027), and cytochrome c oxidase subunit 5b (cluster-1527.419), were found to be up-regulated, presumably to meet energy demands under response to environment stress.

Most of the photosynthesis-related genes in the chloroplasts were down-regulated (Table 2). For instance, the gene encoding photosystem II P680 reaction center D1 protein (PsbA, cluster-1527.39156), a component of the photosynthetic reaction center in PSII, decreased by 2.94 (log_2_)-fold under 1/16 P-limited conditions compared with the control, while the cytochrome b6 gene (PetB, Cluster-1527.15944), which is involved in electron transfer of photosynthetic phosphorylation, decreased by 3.60 (log_2_)-fold. In addition, several genes involved in fatty acid biosynthesis and amino acid metabolism, such as acetyl-CoA carboxylase (luster-1527.38700), malonyl-CoA acyl carrier protein transacylase (FabD, luster-1527.31766), and S-adenosylmethionine synthetase (luster-1527.83262) were up-regulated.

It is worth noting that some members of the ATP-binding cassette (ABC) transports superfamily, including ABCA3 (cluster-1527.71271), ABCB1 (cluster-1527.63439), and ABCC1 (cluster-1527.10718), were observed to be up-regulated under the P-limited conditions. Additionally, some genes involved in cell endocytosis, phagosome formation, and ubiquitin-mediated proteolysis, such as calreticulin (cluster-5481.0), stromal membrane-associated protein (cluster-1527.1659), ras-related proteins (cluster-1527.16544, cluster-1527.86789, cluster-1527.76712, cluster-6525.0), and ubiquitin family proteins (Ccuster-1527.11208), were also up-regulated under the P-limited conditions.

### 3.4. RT-qPCR Validation of Differentially Expressed Genes

To validate the RNA-Seq data in gene expression related to photosynthesis, lipid metabolism, endocytosis, and ABC transporters, RT-qPCR was performed. Five DEGs—photosystem II P680 reaction center D1 protein (PsbA), photosystem II CP43 chlorophyll apoprotein (PsbC), cytochrome b6-f complex subunit 4 (PetD), malonyl coenzyme A-acyl carrier protein transacylase (MCAT), and ATP-binding cassette subfamily B1 (ABCB1) were selected for RT-qPCR analysis. The expression levels of these genes were normalized with the reference gene β-tubulin. As shown in Appendix A, the expression levels of these DEGs between P-replete and 1/16 P-limited treatments as determined by RT-qPCR were basically consistent with those determined by RNA-seq. These results indicate that, overall, the RNA Seq method provides a correct reference for expression profiling analysis.

### 3.5. Lipid and Starch Accumulation under P-Limited Conditions

In the P-replete cells, lipid bodies appear relatively bright and show fewer oil bodies with green fluorescence. In *P. lima* cells from 1/16 P-limited and 1/4 P-limited cultures, lipid bodies appear much darker and show more oil bodies with green fluorescence (Figure 3).

In parallel with CLSM observations, lipid contents in *P. lima* cells were estimated by measuring their fluorescence intensity with a microplate reader. Lipid content in *P. lima* cells grown under 1/16 P-limited and 1/4 P-limited conditions was about 5.35-fold and 1.82-fold higher, respectively, than those in the controls (Figure 4A). In addition, starch contents detected with a starch content assay kit in 1/16 P-limited and 1/4 P-limited cells (62.76 ± 3.58 pg cell^−1^ and 52.93 ± 5.43 pg cell^−1^, respectively) were significantly higher than those in the P-replete control (20.21 ± 3.07 pg cell^−1^) (Figure 4B).

### 3.6. DSP Toxin Content in P. lima under P-Limited Conditions

Through toxin analysis using HPLC–MS/MS, only OA and DTX1 were detected in the *P. lima* cell extract. DSP toxin content in *P. lima* cells shows significant differences between cultures under P-limited and P-replete conditions (Figure 5). The highest OA concentration of 1.87 ± 0.38 pg cell^−1^, found in 1/16 P-limited cultures, was significantly higher than observed in the 1/4 P-limited cultures (1.13 ± 0.33 pg cell^−1^) and 1P-replete groups (0.58 ± 0.15 pg cell^−1^) (Figure 5A). Meanwhile, the highest DTX1 concentration of 22.14 ± 0.73 pg cell^−1^, found in 1/16 P-limited cultures, was significantly higher than observed in the 1/4 P-limited cultures (13.60 ± 1.39 pg cell^−1^) and 1P-replete groups (4.84 ±1.33 pg cell^−1^) (Figure 5B). In addition, the content of DTX is significantly higher than that of OA in our study, indicating this strain of *P. lima* mainly produces DTX.

## 4. Discussion

### 4.1. Carbon Metabolism and Photosynthesis under P-Limited Conditions

Almost all organisms can synthesize pyruvate from glucose or other polysaccharides through glycolysis pathway, providing ATP, reducing power, and key metabolites [4]. In the glycolysis pathway, the up-regulation of pyruvate kinase, phosphoglycerate kinase, pyruvate dehydrogenase, and succinate dehydrogenase (Table 2) was found in the P-limited cells, which would supply the key intermediate pyruvate and could be used as the precursor for acetyl-CoA formation for the TCA cycle, or lipid or amino acid synthesis. The up-regulation of glycolytic activity suggests that *P. lima* requires additional energy to activate various defense mechanisms to cope with P deficiency. These results are consistent with recent studies in respect of *Prorocentrum* spp. [4,19]. Hou et al. reported that the glycolytic activity of *P. lima* was enhanced and several enzymes involved in glycolysis were up-regulated under nitrogen limitation [19]. Zhang et al. found that the expression levels of 6-phosphofructokinase and pyruvate dehydrogenase in *P. donghaiense* were significantly up-regulated under P limitation and down-regulated 28 h after P refeeding [4].

Previous studies have shown that P deficiency could lead to a decrease in photosynthesis and the growth of microalgae as well as plants [2,26,36,37]. *Fv*/*Fm* is the maximum photochemical quantum yield of the photosystem II (PS II) reaction center. As an important chlorophyll fluorescence parameter, *Fv*/*Fm* represents the light energy conversion efficiency of the photosynthetic process [19], as well as being a potential valuable indicator of the photoinhibition and nutrient limitation of phytoplankton [43]. *Ft* is the instantaneous chlorophyll fluorescence yield and represents the emission from the excited chlorophylls in the PSII antenna, which is related to the concentration of chlorophyll [44]. In our study, both *Ft* and *Fv*/*Fm* in *P. lima* decline under P deficiency (Figure 1F,G), a fact which is consistent with similar observations in other planktonic microalgae [4,36]. Moreover, some genes involved in photosynthesis, such as photosystem II P680 reaction center D1 protein (PsbA), photosystem II CP43 chlorophyll apoprotein (PsbC), photosystem II CP47 chlorophyll apoprotein (PsbB), photosystem I P700 chlorophyll a apoprotein A1 (PsaA), photosystem I P700 chlorophyll a apoprotein A2 (PsaB), cytochrome b6 (PetB), cytochrome b6-f complex subunit 4 (PetD), and ferredoxin (PetF), were also down-regulated under the P-limited conditions (Table 2), indicating that P limitation had a severe influence on the photosystems in *P. lima* (Figure 6).

The mechanism of carbon fixation in *P. lima* remains unclear. Based on other studies, *P. lima* may possess a typical C3 or an intermediate C3–C4 photosynthetic pathway [19,45]. In our study, the genes involved in carbon fixation pathways, such as ribulose bisphosphate carboxylase (RbcL, cluster-1527.41437), phosphoglycerate kinase (PGK, cluster-1527.83027), glyceraldehyde-3-phosphate dehydrogenase (GAPA, cluster-1527.43164), and triosephosphate isomerase (TPI, cluster-1527.53219), were found to be increased under the P-limited conditions (Table 2). The up-regulation of these genes suggests that the gluconeogenesis process and starch synthesis are activated. Moreover, the up-regulation of pyruvate orthophosphate dikinase (PpdK, cluster-1527.73147) were also identified. PpdK is a critical enzyme involved in C4 photosynthesis, indicating the possibility of C4 biosynthesis pathway in *P. lima* (Figure 6).

### 4.2. Phosphate Transport, Utilization, and Homeostasis

Dissolved inorganic phosphorus is the preferred P source for phytoplankton [24,25]. Under P-limited conditions, many microalgae can increase phosphate uptake by enhancing the expression of phosphate transporters, such as the green algae *Chlamydomonas reinhardtii* [46], the diatoms *Thalassiosira pseudonana* [29], and the dinoflagellates *Karenia mikimotoi* [28] and *Prorocentrum donghaiense* [32]. In addition, transcripts of several phosphate transporters have been shown to significantly increase by low P treatment in many plant cells [38,39,47]. In this study, numerous transcripts of phosphate transporters, including inorganic phosphate transporters, mitochondrial phosphate transporters sodium-dependent inorganic phosphate transporters, triose-phosphate transporters, sphingosine-1-phosphate transporter, and glycerol-3-phosphate transporter, were identified in *P. lima*. Of these, the expression of some phosphate transporters was significantly up-regulated in the 1/16 P-deficient cells (Table 2), indicating that *P. lima* possesses phosphate uptake systems to achieve adaptation to ambient P deficiency (Figure 6).

When DIP is depleted, DOP can serve as the major alternative P source for microalgae [4,27,29]. Alkaline phosphatase is commonly present in algae and usually enhanced under P limitation conditions. It can hydrolyze phosphate from phosphomonoesters for assimilation by the cell [24]. In the present study, APA in the 1/16 P-limited cells was markedly evaluated compared to that in the P-replete cells (Figure 1E). In addition, transcripts of several other phosphatases, such as acid phosphatase, phospholipase A1, phospholipase A2, and phosphodiesterase were significantly up-regulated in P-deficient cells. Acid phosphatase plays important roles in metabolism such as hydrolysis of phospholipid materials, participation in autophagy processes, and endomembrane recycling [48]. The up-regulation of acid phosphatase could also be observed in many plant transcriptome studies under conditions of phosphorus deficiency, such as peanut [39], quinoa [49], rice [50], and soybean [51]. Phospholipase plays central roles in the maintenance and remodeling of the cell membrane by hydrolyzing membrane phospholipids [52]. Phosphodiesterase can hydrolyze assorted structural phospholipids, producing phosphatidic acid and various head groups [53]. Furthermore, phosphodiesterase plays various roles in membrane transport, cell migration, hormone signaling, and the environmental stress response [54,55]. Accordingly, under P deficiency, the utilization of intracellular phosphor esters was enhanced in *P. lima*, and organic P became the major P source (Figure 6).

Endocytosis plays vital roles in signal transduction, plasma membrane homeostasis, transport of extracellular substances into cells, and the cellular response to environmental stimuli [4,56,57]. In eukaryotic cells, clathrin-mediated endocytosis is the major transport mechanism through which extracellular molecules are packaged into clathrin-coated vesicles and taken up into the cells [4,57]. In our study, some genes such as ADP-ribosylation factor 1/2 (ARF1/2), stromal membrane-associated protein (SMAP), and charged multivesicular body proteins (CHMPs) were enriched in the endocytosis metabolic pathway and the transcript expression levels were significantly up-regulated in the P-limited cells (Table 2). ARF1/2 factor plays pivotal roles in vesicle formation [58]. SMAP is a subfamily of the ARF GTPase-activating proteins and is involved in ARF-mediated vesicular transport [59]. CHMPs are required for the formation of the multivesicular body, a late endosomal structure that fuses with the lysosome to degrade endocytosed proteins [60]. Therefore, endocytosis might be an important strategy for *P. lima* to utilize extracellular nutrients, including DIP and DOP, and to maintain intracellular P homeostasis under P-limited conditions (Figure 6).

### 4.3. Lipid and Starch Accumulation under P-Limited Conditions

It is well-known that the storage of carbohydrate and lipid in many algae play important roles in response to ambient changes such as nutrition concentration [19,61,62,63]. For microalgae, carbon sources usually include polysaccharides and lipids [64,65]. The excess carbon generated by photosynthesis can be used to synthesize storage molecules such as triglycerides or starch. Previous studies have shown that nitrogen and phosphorus deficiency could trigger lipid and starch accumulation in algal cells [19,61,66], but lipid and starch accumulation have not yet been studied for *P. lima* in response to P limitation. Feng et al. reported that the lipid content of *Chlorella zofingiensis* grown in phosphate-deficient media (44.7%) was higher than that grown in full media (33.5%) [67]. Khozin-Goldberg et al. observed that the cellular total lipid content of *Monodus subterraneus* cells increased from 1.6 to 4.4 fg cell^−1^ with decreasing phosphate availability from 175 to 0 μM, which was mainly due to the dramatic increase in triacylglycerol levels [68]. Yao et al. studied cell growth and starch accumulation in *Tetraselmis subcordiformis* under phosphorus deprivation and the maximum starch content of 44.1% was achieved with initial cell concentration of 1.5 × 10^6^ cells mL^−1^ [69].

From our observations, total lipid and starch contents in *P. lima* cells under the 1/16 P-limited conditions were significantly higher than those of P-replete counterparts, at about 5.35-fold and 3.10-fold of the controls, respectively (Figure 3 and Figure 4). Accordingly, several enzymes related to fatty acid synthesis, such as acetyl-CoA carboxylase (cluster-1527.38700), malonyl coenzyme A-acyl carrier protein transacylase (cluster-1527.31766), and long-chain acyl-CoA synthetase (cluster-4976.0) were up-regulated by 2.00 (log_2_)-, 1.16 (log_2_)-, and 3.70 (log_2_)-fold, respectively (Table 2). Some genes involved in starch synthesis, such as soluble starch synthase 1 (cluster-1527.40721) and glucose–starch glucosyltransferase (cluster-1527.11343), were also up-regulated by 0.94 (log_2_)-fold and by 0.57 (log_2_)-fold, respectively. Consequently, these genes may be involved in lipid and starch biosynthesis, and the regulation of intracellular carbon metabolism of *P. lima* during P stress (Figure 6).

Studies have shown that autophagy plays an important role in lipid production and photosynthetic metabolism in microalgae [70]. The plastid membranes and cellular components generated by autophagy might be in favor of recycling nutrients and the accumulation of carbon source. A previous study showed that in *P. lima* cells under nitrogen-limited conditions, the chloroplasts appeared smaller and were less abundant, which suggested the presence of autophagy in *P. lima* and remodeling of the chloroplast membrane to accumulate lipid and starch [19]. Moreover, *P. lima* contains acid phosphatases in lysosomes, which have a catabolic function, contributing to food digestion and autophagy process [19]. These enhance the possibility of autophagy in response to P-deficiency in *P. lima* (Figure 6). This effect of P-limitation on autophagy was also supported by our study, where some genes involved in the ubiquitin pathway and autophagy process were markedly up-regulated (Table 2). As is well-known, the ubiquitin pathway is necessary for the extensive degradation of cellular proteins and the targeted hydrolysis of specific regulatory proteins in eukaryotic cells. It appears that ubiquitin can promote the degradation of a large amount of protein used for energy recovery under phosphorus-deficient conditions, which may contribute to the accumulation of carbon sources (Figure 6). Starch and lipid contents were found to be increased in the P-limited cells, which provided further evidence that P-limitation could induce the accumulation of lipid and starch in *P. lima* through recycling plastid membranes or cellular components via autophagy and the ubiquitin pathway.

### 4.4. DSP Toxins Production in P. lima

Previous studies showed that the culture conditions, such as nutrients, salinity, and temperature, could induce significant changes in DSP toxin levels of *P. lima* [71,72,73,74,75]. Of these, starvation of phosphorus could increase the production of toxins [17,21,22]. Vanucci et al. reported that OA content in *P. lima* significantly increased 2.3-fold compared to the control under P-limitation conditions [21]. Hou et al. found that the OA contents per cell under P-limitation conditions increased 2.01-fold with respect to control [22]. In line with these studies, the content of OA and DTX1 in our study were found to increase under 1/16 P-limited conditions, significantly higher than control counterparts by 3.2-fold and 4.6-fold, respectively (Figure 5). The production of toxins may be a compensatory competitive strategy for dinoflagellate under low nutrient conditions. It is worth noting that cell growth and photosynthetic efficiency were down-regulated under P-limitation, but toxin cellular contents were up-regulated, which showed the strong negative correlations between toxin production and cell growth (or photosynthesis).

A previous study suggested that ATP-binding cassette (ABC) transporters might be responsible for the transport or efflux of DSP toxin in *P. lima* [19]. ABC transporters are composed of a large and diverse superfamily of transmembrane proteins that participate in the active transport of various substrates in prokaryotes and eukaryotes. To date, a large number of ABC transporters have been identified in various species, such as multidrug resistance (MDR or ABCB), multidrug-resistance-associated protein (MRP or ABCC), and pleiotropic drug resistance (PDR or ABCG), which are involved in the membrane transport of endogenous secondary metabolites, playing an important role in the avoidance of self-toxicity [76]. DSP toxin content and MRP activity were found to be increased conformably over time under P-limited conditions compared with control counterparts [22]. In our study, the OA and DTX1 contents per cell under P-limitation conditions increased significantly with respect to the control (Figure 5). Accordingly, the expression of ABCA3 (cluster-1527.71271), ABCB1 (cluster-1527.63439), and ABCC1 (cluster-1527.10718) was found to be increased by 1.35 (log_2_)-fold, 1.81 (log_2_)-fold, and 1.64 (log_2_)-fold, respectively (Table 2). This consistency further indicates that ABC transporters might be related to the transport or efflux of DSP toxins in *P. lima* (Figure 6).

## 5. Conclusions

This study investigated the molecular mechanisms expressed by *P. lima* to respond to ambient P deficiency through RNA sequencing and physiological analysis. A total of 148 unigenes were significantly up-regulated, and 30 unigenes were down-regulated under 1/4 P-limited conditions, while 2708 unigenes were significantly up-regulated, and 284 unigenes were down-regulated under 1/16 P-limited conditions. The expression levels of genes encoding P transporters and organic P utilization in *P. lima* were significantly up-regulated, indicating the strong ability of *P. lima* to utilize DIP as well as DOP. Cell membrane phospholipids or other phosphate ester compounds could serve as a major P source for P-deficient cells, while the enhancement of endocytosis and autophagy would allow the cells to capture more DOP and regulate P homeostasis. The biochemical and physiological analyses reveal that starch, lipid, and toxin contents increase, while photosynthetic efficiency decreases under P deficiency. The accumulation of lipid and starch in P-deficient cells could provide an energy source for *P. lima* to cope with adverse environmental conditions, such as a sudden exposure to P stress. Meanwhile, the up-regulation of DSP toxins may be a compensatory competitive strategy of *P. lima* subject to P stress, which can be used as a deterrent to predators feeding on planktonic algae. Over all, our results indicate that *P. lima* has evolved diverse adaptive strategies to acclimatize to low P environments and storage of carbon sources. Our study provides a novel insight into the unique transcriptomic response of *P. lima* under P deprivation. We believe that our study would make a significant contribution for the researchers who study the mechanism of harmful algal bloom formation and regulation of toxin biosynthesis, as well as environmental monitoring.

## Figures and Tables

**Figure 1 microorganisms-11-02216-f001:**
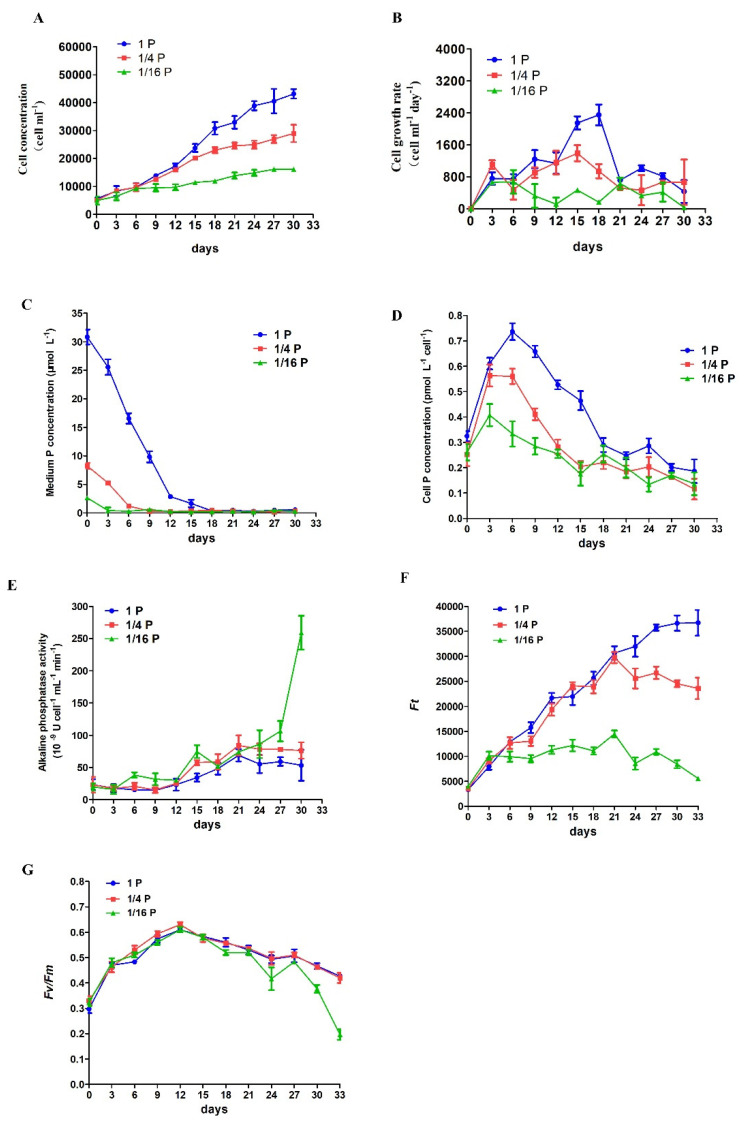
Physiological responses of *Prorocentrum lima* to P limitation. (**A**) Cell concentration; (**B**) cell growth rate; (**C**) extracellular P concentration; (**D**) cellular P concentration; (**E**) alkaline phosphatase activity (APA); (**F**) instantaneous chlorophyll fluorescence (*Ft*); (**G**) photochemical efficiency of photosystem II (*Fv*/*Fm*). Cells were cultured in three replicates for each treatment and samples collected every three days for enumeration.

**Figure 2 microorganisms-11-02216-f002:**
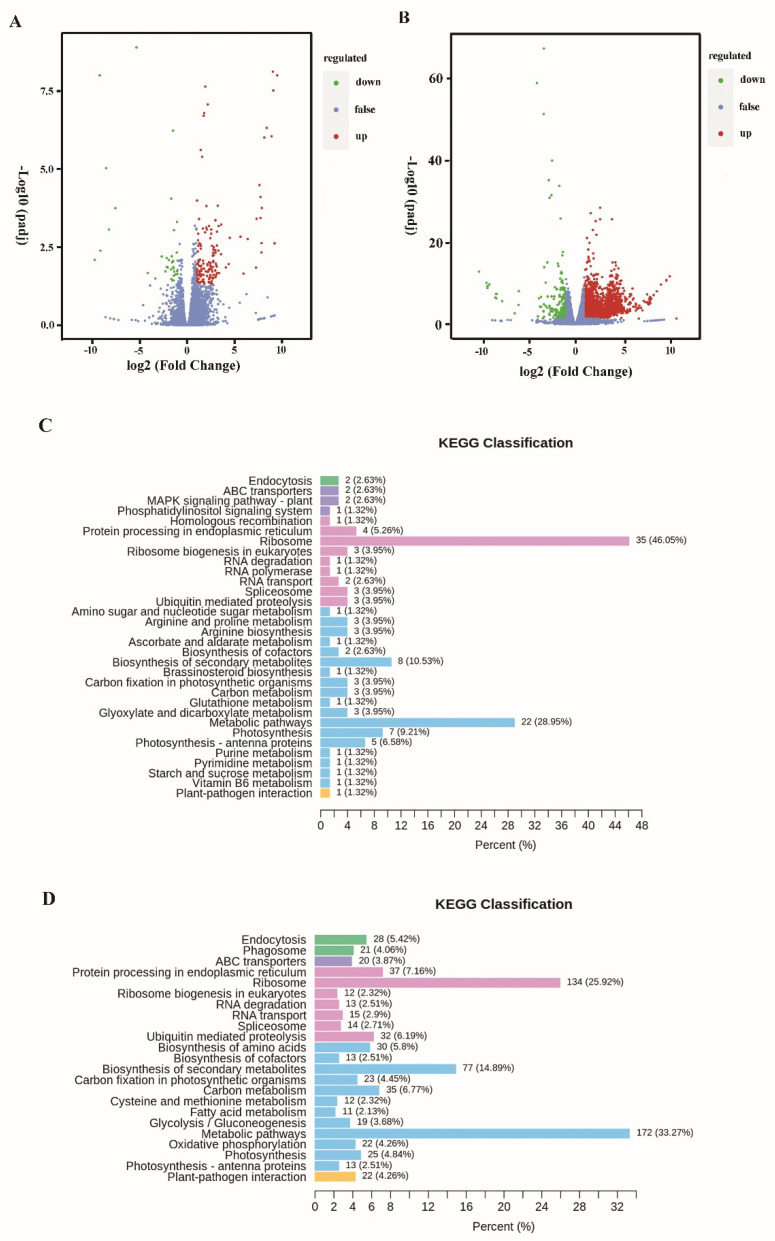
Differential expression analysis in *P. lima* under different levels of P-limitation and a P-replete control. (**A**) Differential gene volcano map between 1/4 P-limited and P-replete conditions. (**B**) Differential gene volcano map between 1/16 P-limited and P-replete conditions. The horizontal axis represents the multiple changes in gene expression, while the vertical axis represents the significance level of differential genes (log_2_ fold change). The red dots represent up-regulated genes, the green dots represent down-regulated genes, and the blue dots represent insignificantly altered genes. (**C**) KEGG classification between 1/4 P-limited and P-replete conditions. (**D**) KEGG classification between 1/16 P-limited and P-replete conditions. The horizontal axis represents the proportion of the number of differentially expressed genes annotated to this pathway to the number of differentially expressed genes annotated, while the vertical axis represents the name of the KEGG pathway.

**Figure 3 microorganisms-11-02216-f003:**
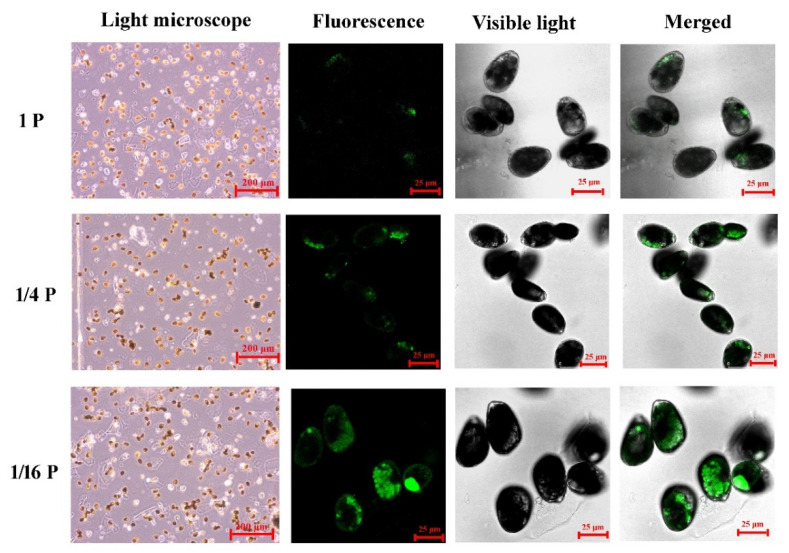
Light microscopy (LM) and confocal microscopy (CLSM) images of *P. lima* cells in P-limited conditions and a control. Representative confocal microscope images of *P. lima* cells showing oil bodies with green fluorescence are displayed.

**Figure 4 microorganisms-11-02216-f004:**
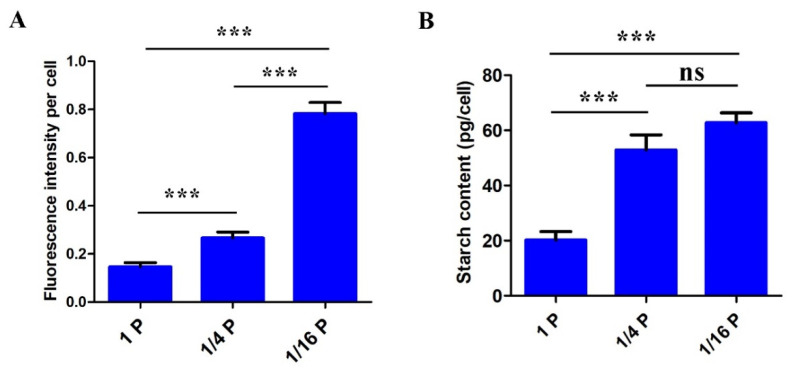
Lipid (**A**) and starch (**B**) content in *P. lima* cells under two P-limitation treatments and a control. (mean ± SD, *n* = 3). Differences between the treatment groups are indicated by *** (*p* < 0.001) or ns (non-significant difference).

**Figure 5 microorganisms-11-02216-f005:**
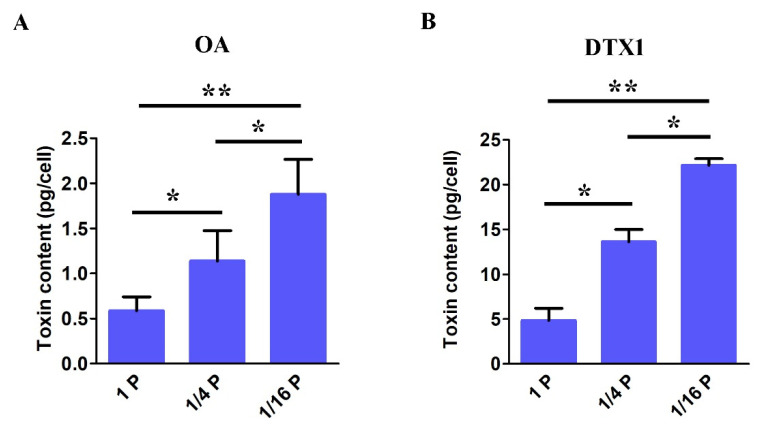
DSP toxin content of OA (**A**) and DTX1 (**B**) in *P. lima* cells under P-replete and P-limited conditions. (Mean ± SD, *n* = 3). Significant difference between the treatment groups is indicated by * (*p* < 0.05), ** (*p* < 0.01).

**Figure 6 microorganisms-11-02216-f006:**
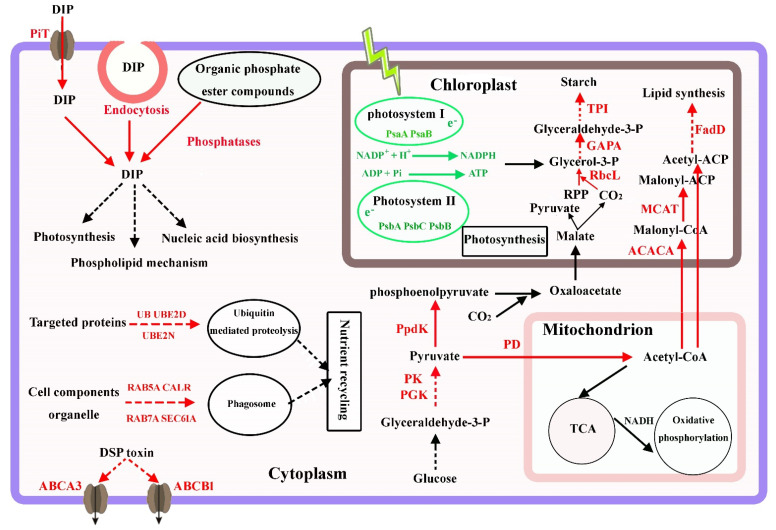
Phosphorus utilization strategies and proposed general transcriptional changes in *P. lima* under P deficiency conditions. Changes in the transcript of abundance of associated genes are indicated with different colors where red means up-regulation and green means down-regulation. DIP: dissolved inorganic phosphorus; PiT: inorganic phosphate transporter; RPP: ribulose-1.5-bisphosphate; TCA: tricarboxylic acid cycle. The other genes with abbreviations can be seen in the Table 2).

**Table 1 microorganisms-11-02216-t001:** Transcriptome assembly statistics and annotations of *P. lima* gene catalogue.

Sample	Raw Reads	Clean Reads	GC Content (%)	Q20 Percentage (%)	Number of Unigenes	BLASTx Analysis Percentage (%)						
						KEGG	NR	Swiss-Prot	TrEMBL	KOG	GO	Pfam
1P	110,831,089	81,618,516	58.86	97.48	88,186	15.63	22.46	11.73	45.91	15.02	35.42	43.44
1/4P	110,497,380	83,366,479	58.87	97.57	90,706	17.97	24.61	14.19	47.42	17.38	37.21	45.00
1/16P	115,006,749	90,073,946	58.54	97.71	91,890	19.03	25.58	15.30	48.09	18.45	38.02	45.72
Average Data	112,111,739	85,019,647	58.76	97.59	90,261	17.54	24.22	13.74	47.14	16.95	36.88	44.72

**Table 2 microorganisms-11-02216-t002:** Representative differentially expressed genes in *P. lima* under P-limited conditions.

Functional Classification	Gene	Description	Unigene ID	log_2_ (Fold Change) (1/16P vs. 1P)
Ribosome	*RP-S20e*	Small subunit ribosomal protein S20e	Cluster-1527.83547	3.92
	*RP-S11e*	Small subunit ribosomal protein S11e	Cluster-1527.1298	3.56
	*RP-L34e*	Large subunit ribosomal protein L34e	Cluster-1527.8476	3.77
	*RP-S5e*	Small subunit ribosomal protein S5e	Cluster-1527.76135	3.65
	*RP-L10e*	Large subunit ribosomal protein L10e	Cluster-5599.0	7.70
Stress–shock proteins	*Hsp 70*	Heat shock 70 kDa protein, mitochondrial-like	Cluster-1527.8250	4.34
	*Hsp 90*	Heat shock protein 90	Cluster-1527.77292	5.04
	*Hsp 83*	Heat shock protein 83	Cluster-1527.9757	4.57
	*HSPA1s*	Heat shock 70 kDa protein 1/2/6/8	Cluster-1527.86746	4.97
Photosynthesis system	*PsbA*	Photosystem II P680 reaction center D1 protein	Cluster-1527.39156	−2.94
	*PsbC*	Photosystem II CP43 chlorophyll apoprotein	Cluster-1527.45732	−3.30
	*PsbB*	Photosystem II CP47 chlorophyll apoprotein	Cluster-1527.50539	−2.51
	*PsaA*	Photosystem I P700 chlorophyll a apoprotein A1	Cluster-1527.41047	−2.44
	*PsaB*	Photosystem I P700 chlorophyll a apoprotein A2	Cluster-1527.52290	−2.79
	*PetB*	Cytochrome b6	Cluster-1527.15944	−3.60
	*PetD*	Cytochrome b6-f complex subunit 4	Cluster-1527.82828	−3.59
	*PetF*	Ferredoxin	Cluster-1527.21762	−8.44
Carbon fixation	*PpdK*	Pyruvate orthophosphate dikinase	Cluster-1527.73147	1.25
	*RbcL*	Ribulose bisphosphate carboxylase	Cluster-1527.41437	1.95
	*GAPA*	Glyceraldehyde-3-phosphate dehydrogenase	Cluster-1527.43164	2.54
	*TPI*	Triosephosphate isomerase	Cluster-1527.53219	1.05
Phagosome	*RAB5A*	Ras-related protein Rab-5A	Cluster-1527.16544	5.36
	*CALR*	Calreticulin	Cluster-5481.0	4.07
	*RAB7A*	Ras-related protein Rab-7A	Cluster-1527.86789	4.29
	*SEC61A*	Protein transport protein SEC61 subunit alpha	Cluster-5772.0	3.57
	*RAC1*	Ras-related C3 botulinum toxin substrate 1	Cluster-1527.76712	3.78
Endocytosis	*ARF1/2*	ADP-ribosylation factor 1/2	Cluster-1527.8157	4.11
	*SMAP*	Stromal membrane-associated protein	Cluster-1527.1659	4.74
	*HSPA1s*	Heat shock 70 kDa protein 1/2/6/8	Cluster-1527.86746	4.97
	*RAB11A*	Ras-related protein Rab-11A	Cluster-6525.0	4.52
	*CHMP4A/B*	Charged multivesicular body protein 4A/B	Cluster-9796.0	4.04
	*CHMP3*	Charged multivesicular body protein 3	Cluster-6541.0	4.52
Glycolysis, oxidative phosphorylation	*PK*	Pyruvate kinase	Cluster-1527.60580	0.91
	*PD*	Pyruvate dehydrogenase	Cluster-1527.62164	1.46
	*PGK*	Phosphoglycerate kinase	Cluster-1527.83027	4.04
	*COX5B*	Cytochrome c oxidase subunit 5b	Cluster-1527.419	4.42
	*SDHA*	Succinate dehydrogenase (ubiquinone) Flavoprotein subunit	Cluster-1527.79011	4.92
	*ATP5B*	F-type H+-transporting ATPase subunit beta	Cluster-1527.10945	4.34
ABC transporters	*ABCA3*	ATP-binding cassette, subfamily A (ABC1), member 3	Cluster-1527.71271	1.35
	*ABCB1*	ATP-binding cassette, subfamily B (MDR/TAP), member 1	Cluster-1527.63439	1.81
	*ABCC1*	ATP-binding cassette, subfamily C (CFTR/MRP), member 1	Cluster-1527.10718	1.64
phosphate transporters	*MPT*	Mitochondrial phosphate transporter	Cluster-1527.80953	3.72
	*TPT*	Triose-phosphate transporter family	Cluster-1527.55083	1.41
Fatty acid biosynthesis	*ACACA*	Acetyl-CoA carboxylase/biotin carboxylase 1	Cluster-1527.38700	2.00
	*MCAT*	Malonyl coenzyme A-acyl carrier protein transacylase	Cluster-1527.31766	1.16
	*FadD*	Long-chain acyl-CoA synthetase	Cluster-4976.0	3.70
Ubiquitin-mediated proteolysis	*UB*	Ubiquitin family protein	Cluster-1527.11208	4.41
	*UBE2D*	Ubiquitin-conjugating enzyme E2 D	Cluster-1527.86049	4.35
	*UBE2N*	Ubiquitin-conjugating enzyme E2 N	Cluster-1527.9310	4.67

## Data Availability

All the raw sequencing reads were deposited in the Sequence Read Archive under Bio Projects PRJNA966759.

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
