# Peer review of "Transcriptomic Analysis of the Response of the Toxic Dinoflagellate Prorocentrum lima to Phosphorous Limitation"

_microorganisms, 2023, doi:10.3390/microorganisms11092216_

Round 1

Reviewer 1 Report

Phosphorus (P) is a crucial nutrient essential for growth and it is a limiting macronutrient for dinoflagellate growth in the ocean. In this study authors compared the RNA-Seq data to global gene expression patterns of a marine diarrheic shellfish poisoning (DSP) toxin-producing dinoflagellate Prorocentrum lima under various level of P. The results showed that genes encoding P transporters, organic P utilization, and endocytosis were significantly up-regulated in the P-deficient cells

Overall, the results suggested that P. lima evolved to adapt to low P conditions.

Some comments to improve the MS:

1.       Please include statistical information on the genes and P-level impact in abstract and conclusions.

2.       Previously transcriptome studies have been used to identify key genes involved in several plants. Authors are suggested to expand the introduction and broaden latest studies in introduction and discussion (See PMID: 36987099).

3.       Table 3, could be moved to supplementary information.

4.       Please provide key implications of present study.

5.       Figure 2A and 2B, increase the font size and enlarge drawings in this figure.

Not applicable.

Reviewer 2 Report

This study explores with conventional methods (growth curves, accumulation of toxins and reserve substances) and with a transcriptomic approach, the physiological response (acclimation) of the DSP toxin producer dinoflagellate P lima subject to two different P-limited treatments. 

The experiments were correctly designed with three replica per treatment plus the control. The application of transcriptomics allows to explore in depth the metabolic pathways involved in the cells’ response to environmental stress. The topic and the approach are of current interest to understand mechanisms of harmful algae development and toxin accumulation.

There are several aspects in the analysis of results and their interpretation that can be improved with little effort. For example, the authors invested efforts in cell counting and measurements to get the growth curves and photosynthetic activity of their P ima cultures. The curves were plotted, but estimates of the division rate parameter (µ), or even the toxin production rate (toxin per cell per day), were not even mentioned. It is convenient to parameterize growth. This will allow a quantitative comparison between treatments.

The authors confuse “toxin production” with toxin content. Toxin production is a dynamic parameter, related with time. Otherwise, they may estimate the toxin content per ml multiplying cell concentration by the toxin content per cell and plot that to get a more realistic idea of production.  Cells may have a high content of toxins when there is uncoupling between growth (cell divisions) and toxin production; .e., you may have a high toxin content (pg OA · cell-1) and little or no production (fg toxin/cell/day or ng toxin/ml/day). Cells continue producing toxin but do no divide, therefore, accumulation per cell increases. What the authors have measured here is the toxin content per cell (not the production). Furthermore, you should mention that you measured particulate (or intracellular) toxin content, but did not measure extracellular toxins (toxins released by the cells in the water).

In summary, transcriptomic analyses and interpretations of metabolic pathways are fine, but if growth curves are plotted, rates should be estimated. This would allow the authors to find out if higher toxin per cell was due to lower or even ceased (stationary phase) division. All these, because toxin production is closely related to division rate and photosynthesis activity.

The English composition needs a thorough editing. I have made extensive, but not fully edited, annotations/corrections. 

DETAILED COMMENTS

Title

The word “transcriptomic” should appear in the title. Not clear what the authors mean by “systematic analysis”. Suggested title: “Transcriptomic analysis of the response (or the acclimation) of the toxic dinoflagellate Prorocentrum lima to phosphorous limitation (or stress)

Material and Methods  

Sections 2.2. and 2.3 . The authors should include equations to estimate growth rate and other parameters with quantifiable measurements.

equation for growth rate (cell µ):

µ = (Ln Nf – Ln N)/tf-to   

where  µ (d-1) is the specific growth rate;   N0 and Nf are the initial and final concentrations of cells/ml, and tf-to is the time interval from day 0 to day f

equation for particulate toxin production rate:

toxin-µ = (Ln Tf – Ln T0)/tf-t0       

where toxin-µ (d-1) is the rate of toxin production); To and Tf are the concentration of toxin per ml of culture (ng/ml) on days 0 and f respectively, and tf-to is the time interval from day 0 to day f; T0 and Tf are calculated by multiplying toxin per cell by te concentration of cells per ml.

Info on these equations/estimates to be found in:

·         Nielsen LT, Krock B, Hansen PJ (2013) Production and excretion of okadaic acid, pectenotoxin-2 and a novel dinophysistoxin from the DSP-causing marine dinoflagellate Dinophysis acuta – Effects of light, food availability and growth phase. Harmful Algae 23:34-45

·         P Rial, M Sixto, JA Vázquez, et al. 2023. Interaction between temperature and salinity stress on the physiology of Dinophysis spp. and Alexandrium minutum: implications for niche range and blooming patterns. Aquatic Microbial Ecology 89, 1-22

DTX1 was the only toxin standard purchased? Why?

Explain the way cells were counted. Which kind of counting chambers and how many cells counted (sample size)?

Results & Discussion

3.1 Physiological Responses

In this section, growth rate estimates should be given as well as toxin production rate (if estimated).

If toxin per ml is estimated, toxin production rate can be estimated easily with an equation similar to that for growth rate

Estimates of cell densities which are averaged (highlighted in yellow) should be given together with their + standard deviation. If they are the maximum reached, they can be called cell maximum” or “final yield”

Figure 1A: cell density or cell concentration are both correct but use the same all through the document and figures

Figure 2 A-B: labelling words in the axes are pale and too small.

Sections 3.5 and 3.6  No need to repeat in Results the methods used (yellow highlight) which were already mentioned in Material & Methods

DISCUSSION

Consider to include discussion on: i)  results maybe showing an imbalance between growth and toxin production rates; ii) relation between toxin production and phtosynthesis

This manuscript is written in comprehensible English but with a bad English language composition and some gramatical errors. Very frequently the sequence of words in a sentence are in an incorrect order. Articles are sometimes missing, but most of the times are redundant (i.e. added without being necessary).

There are also a few cases where the authors are not using the appropriate scientific term for a parameter or property being measured  

Reviewer 3 Report

The manuscript describes the metabolic mechanisms under phosphorus limitation in the marine which is a good topic and falls in the topic of the journal, however, there are some issues to be addressed. The comments are listed below:

1. The English of the text should be checked

2. The authors must be included new, relevant, and more information about other methods. Diverse studies are growing attention for diverse uses as reported by the Awual group according to ScienceDirect. The authors need to indicate such points for a broad range of readers. Moreover, the authors need to cite high-impact articles to make the manuscript high-level. The following specific articles may take be noted in the revision stage of https://doi.org/10.1016/j.jclepro.2019.04.325; https://doi.org/10.1016/j.cej.2017.08.037; https://doi.org/10.1016/j.watres.2011.06.009

3. Comparison between the obtained results and measured in this study with other reported studies should be done and included for more clarity (indicate values not just the number of references).

4. A schematic mechanism describing the adsorption process must be indicated and included (reactions, interactions, etc.)

5. Correct the References using the guide of the Journal. More Conclusions must be included with the best results, and values obtained
